# The Burden of Caring: An Exploratory Study of the Older Persons Caring for Adult Children with AIDS-Related Illnesses in Rural Communities in South Africa

**DOI:** 10.3390/ijerph16173162

**Published:** 2019-08-30

**Authors:** Makhosazane Ntuli, Sphiwe Madiba

**Affiliations:** Department of Public Health, Sefako Makgatho Health Sciences University, Pretoria 0001, South Africa

**Keywords:** HIV and AIDS, older persons, South Africa, caregiving role, poor resources, rural communities

## Abstract

Since the start of the HIV and AIDS epidemic, very little research has been conducted on the older persons’ provision of HIV-related care to adult children. This is despite the fact that a large proportion of adults who die of AIDS-related illnesses stay with their elderly parents during the terminal stage of their illnesses. This paper explores how older persons in rural settings experience caring for their adult children with AIDS-related illnesses. In-depth interviews took place with older persons aged 60 years and above. The qualitative data analysis was informed by thematic approach to identify and report themes using inductive approach. The paper found that the older persons undertake the caring role without resources and support. As a result, they are burdened with having to care for adult children with AIDS-related illness. Fatigue arising from the hard work of physically caring for their sick adult children day and night adds to the physical burden on the older persons. Older persons will continue to carry the burden of caring for people with AIDS-related illnesses due to the increase in the number of new infections in South Africa. There is a need to involve them in HIV/AIDS programmes; their experience and wisdom would surely contribute positively and assist in addressing HIV prevention.

## 1. Introduction

Sub-Saharan Africa has the most virulent HIV/AIDS epidemic in the world and accounts for about 70% of the global number of 25 million infected people. South Africa has the highest prevalence of people living with HIV (6.1 million) in the world and accounts for 16% of the world’s new infections [1]. In 2016, the African region had 800,000 new infections, and South Africa accounted for one-third (270,000) of the region’s new infections [2]. According to UNAIDS [1], AIDS-related deaths have been reduced by more than 51% since their peak in 2004. In the African region, although about 66% of the adults living with HIV have been put on antiretroviral treatment (ART), there were 380,000 AIDS-related deaths in 2017 [2]. HIV and AIDS affect older people in a number of ways. Since the beginning of the HIV/AIDS epidemic, many older persons have lost their adult children to AIDS, and the phenomenon continues to exist [3].

Although the administration of ART in many countries in sub-Saharan Africa (SSA) has increased substantially, the burden of caregiving has not changed. In most of these countries, the healthcare services offer significant but limited care and support to those with AIDS, leaving the bulk of the work to their families [4]. Consequently, older persons assume the important role of providing care and support to their adult children when they become debilitated by AIDS-related illnesses [5].

Generally, kinship caregiving and family responsibility for the sick in SSA are the responsibilities of women. In these societies, many caregivers are older women, aunts and grandmothers to those that are sick [6,7,8]. At the beginning of the HIV epidemic, this caring role was automatically transferred to females, both old and young. However, the high burden of care for people living with HIV/AIDS fell mainly on the elderly women who cared for their adult children as well as their orphaned grandchildren [9]. Women take the responsibility for the care of their sick adult children, as it has been a long-standing societal expectation that they would do so [8,10,11]. Earlier, in the beginning of the HIV/AIDS epidemic, the majority of the seriously ill who lived and worked away from home returned to their rural homes, where the social networks and support systems were stronger, to recover and or possibly to die [12]. Thus, Saengtienchai and Knodel believe that adult children with advanced AIDS-related illnesses return to their parents when they can no longer care for themselves [13].

In South Africa, 85% of the people above 60 years old in some communities live in multigenerational households. In such a household, the elders stay with individuals of various ages, ranging from adults to young children [14]. Commonly, these households include unemployed adult children who may or may not have their own children and live with their elderly parents for economic benefit [15]. This is a common occurrence in many poor societies in South Africa, particularly in rural areas, where older persons are faced with a complex, high rate of unemployment of their adult children [15]. In the era of the HIV epidemic, this multigenerational living arrangement has resulted in older people taking responsibility for providing care to their adult children with AIDS [9,16,17]. Although older persons are compelled by circumstances rather than by choice to assume their caregiving roles, they feel a strong sense of responsibility to be care providers as the heads of their families. Thus, they reckon that the responsibility of being care providers naturally rests on them [18,19,20].

Since the start of the HIV/AIDS epidemic, very little research has been conducted on the older persons affected by the AIDS-related illnesses and deaths of their adult children. Thus, little is known about older persons’ provision of HIV-related care to adult children in SSA (Small et al., 2017). This is despite the fact that a large proportion of adults who die of AIDS-related illnesses stay with their elderly parents during the terminal stage of their illnesses [3,5,21]. Furthermore, it is projected in Statistics South Africa (STATS SA) [22,23,24] that the population ratio of people over the age of 60 years will nearly double from 7% to 13% from 2006 to 2050. As already stated, society puts demands and expectations on the elderly to care for sick people, but there are limited current data in South Africa on the how the process of caring affects older persons’ physical and emotional wellbeing. Furthermore, the literature has focused largely on the impact of caring for orphans and vulnerable children rather than for both adult children and grandchildren [25]. Therefore, this paper explores how older persons in rural settings experience caring for their adult children with AIDS-related illnesses. The elderly have been largely ignored with respect both to how they are affected and to how they contribute to fighting the HIV epidemic [4]. Providing care to an ill adult child who is inevitably facing imminent death is an extremely trying and difficult experience.

## 2. Methodology

### 2.1. Study Design

The data for this paper were extracted from a broader study conducted in the sub-District of Nkangala in Mpumalanga Province. The paper emanates from the first author’s doctoral study in Public Health (DrPH), which used a mixed-method design. The purpose of the study was to develop an HIV and AIDS educational programme for elderly parents. The quantitative data were used to establish the demography of the older persons and the extent of their knowledge of HIV/AIDS, as well as their attitudes towards HIV/AIDS. This better understanding of the level of knowledge older persons about HIV/AIDS informed the development of an educational programme to meet their needs as they care for their sick family members. The qualitative data generated in the study were used to explore the caring role of the older persons and the challenges they faced, to develop an understanding that was the basis of the development of a relevant educational programme. Integrating elements from both the quantitative and the qualitative data brought about an understanding of the needs of the older persons who have children/family members with HIV or who are sick with AIDS-related illnesses. The data sets were integrated after they had been analysed separately and with equal emphasis [26,27].

### 2.2. Setting and Participants

The study site was primary health facilities in one of the six subdistricts of Nkangala District in Mpumalanga Province, South Africa. The district is approximately 75 km northeast of Pretoria and is predominantly rural, with 57 villages and adjacent farms. The subdistricts are characterised by poverty, poor education, unemployment and limited infrastructure but have adequate access to PHC services, as most people walk to the local PHC facility. The proximity of the subdistrict to Gauteng Province accounts for its having a greater population than the other subdistricts, due to the return of people from the urban to this rural area to avoid expensive housing, rates and taxes in the urban area. The subdistrict has 7 community centres, 13 clinics and a level 1 hospital, and the area is serviced by 9 ward-based outreach teams. Particular facilities were selected for the study because of the numbers of elderly people that attend them and collect chronic medication.

The research population consisted of persons aged 60 years and above that presented at the health facilities to collect chronic medication; none of the participants who were selected were critically ill, but they were of sound mind. Those who were to participate in face-to-face interviews were selected, using purposive sampling, from the larger group of older persons who had been sampled for the cross-sectional survey. They were asked if they were taking care of or had taken care of an adult child with an AIDS-related illness, and if the response was positive, the participant was requested to participate in the in-depth interview. They were recruited from the twelve health facilities selected for this purpose. The number of persons recruited for the interviews, 31 in all, was guided by data saturation.

### 2.3. Data Collection

Data were collected between June and August 2016 by the first author, who was assisted by experienced and trained research assistants. Face-to-face in-depth interviews took place in the selected facilities with the 31 participants, using an unstructured interview guide. Although the research assistants were experienced in interviewing, the second author trained them in a one-day training session on the objectives of the study and in handling sensitive information, such as caring for family members with AIDS-related illness. The interview schedule consisted of open-ended questions covering topics such as their understanding of HIV and AIDS, the duration of the care provided or the illness, the activities performed, the circumstances leading to the participant’s having to care for an ill adult child, their thoughts about disclosing the child’s HIV status, the secrecy of the HIV diagnosis, experience of rejections from family members and or community, their opinions on how to deal with discrimination, the support received and the envisaged future support.

The interviews were conducted in the local language in a room separate from the main facilities of the clinic, to ensure privacy. They were audio recorded and each lasted for about 60 min. Ethical clearance for this study was obtained from the Research and Ethical Committee of the School Research Ethics Committee, the Sefako Makgatho Health Sciences University Research Ethics Committee and, finally, from the Mpumalanga Provincial Department of Health and the Nkangala Health District. Written informed consent was obtained from the prospective participants before the interviews. They were also informed that participation in the interviews was voluntary and were assured about confidentiality and anonymity.

### 2.4. Data Analyses

To analyse the qualitative data, the authors adopted thematic analysis approach to identify and report patterns or themes within data [28] using an inductive approach. The first author was assisted in transcribing the audio recordings of the interviews in IsiZulu by the research assistants who conducted the interviews. Each transcript was then translated into English. Analysis first began by careful reading of the transcripts and identifying initial codes that emerged from the data. These codes were used to develop a codebook, and the emerging themes were discussed with the co-author until consensus had been reached about their definitions and the finalisation of the codebook. NVivo 10, which is qualitative data analysis software (QSR International, Melbourne, Australia), was used to code the transcripts. The codes were then grouped into categories, and emerging themes across the transcripts were identified. Final themes were decided upon by agreement between the authors. Descriptive statistics for the quantitative data were analysed using STATA version 14.

To increase the rigour and the credibility of the study findings, all the authors were involved in the analysis and interpretation of the data. Trustworthiness was established through credibility, transferability, dependability and conformability [29]. We used several strategies and methodologies such as probing, peer debriefing, triangulation, keeping an audit trail and transcribing verbatim to ensure rigour.

## 3. Findings

### 3.1. Description of the Study Sample

Table 1 presents the socio-demographics of the older persons. The sample consisted of 31 older persons who had cared for adult children with AIDS-related illnesses; all but one were female. The ages of the participants ranged between 62 and 82 years, most (19) of them were aged between 62 and 69 years and almost half (15) had no formal schooling. The older persons were all from multigenerational households. Five of them were living with 9 to 12 household members, 19 had 8 children and only 1 was living alone at the time of data collection. The duration of care ranged from 6 months to over 24 months. The mean age of the sick adults was 33.3 years, range 20–52 years. Almost all the participants (27) were on treatment for hypertension, and they visited the clinic to collect their chronic medication. Twelve of those with hypertension were also talking medication for diabetes mellitus. Most (19) participants reported that their health status was poor and that caring for their adult children affected their health. Twenty-four of the adult children that the participants cared for died (Table 1).

### 3.2. Themes

Six main themes emerged from the analysis of the in-depth interviews (Table 2). The main themes include: (1) learning about the HIV status of the sick child, (2) perceptions about disclosure, (3) the challenges to care, (4) the risk of HIV infection, (5) HIV knowledge and (6) perceived role in the HIV prevention. This paper focusses on one theme, namely, the challenges to care highlight that older persons experience caring as a burden, as is evident in the following subthemes gathered from their responses to explore their caring role and the challenges they faced. The following subthemes: physical effects of providing care, the emotional toll on them, their sense of fulfilment, receiving support for care and limited resources illuminate the burden of care as experienced by the older persons. Although stigma and discrimination are not a subtheme for challenges to care, they are also discussed as they helped to explain the burden of care.

#### 3.2.1. The Physical Effects of Providing Care

The older persons performed labour-intensive tasks such as turning the sick adult children in bed and supporting them to the toilet, which in many instances was outside the house. These tasks were mostly performed without the assistance of other family members. Such heavy physical tasks drained the energy of the already weak older persons, who were getting minimal sleep at night and no rest during the day. Most ended up being physically ill.


*“My health was not fine anymore as after the funeral I came to see the doctor, and the doctor said I have a heart problem.”*
*(69-year-old)*

The reduced physical strength resulted in their inability to look after their households and affected their caring role.


*“Hey, I do not know because I was always sick; I had no energy to pick him up.”*
*(69-year-old)*

Although the participants got sick from caring for their children, they showed commitment in taking care of them and took this to be their sole responsibility.


*“I had to help him …, he was my child, and I had no choice. Even when he spoilt on himself I would wash his clothing and linen, I never slept day and night … Eyi …, it was extremely difficult.”*
*(66-year-old)*


*No there was no one and especially when he had running stomach I was the one who was helping him.”*
*(72-year-old)*

The participants described their experiences of HIV and AIDS as traumatic. They had terrible and painful experiences of HIV and AIDS and perceived it as a deadly disease that had robbed them of their adult children.


*“HIV is deadly. HIV is not good, and many people have died. The young people have died; there are no young men to call upon for help. We cannot even call upon young girls from the neighbourhood to come and help; there is no one.”*
*(74-year-old)*

Some of the participants had more than one adult child sick at a time to care for, either under one roof or in different places.


*“As their elderly parent I saw them getting sick. The other one I saw her coughing and getting skinny and always sleeping with no energy.”*
*(65-year-old)*

#### 3.2.2. Endurance

AIDS is a debilitating disease where the patients’ strength is severely reduced, causing them to lose the ability to take care of themselves. As the disease advances, the patient becomes weaker and more dependent on caregivers. The older persons were the ones who cared for their sick adult children. They performed the caring activities as their parental responsibility towards their adult children. This varied from prolonged months or years as well as short periods of days or weeks of illness.


*“No there was no one, his younger brother was working and all the other children were too afraid to help him. Only my last-born child had the guts to help me with taking care of his older brother. I had to help him …, he was my child, and I had no choice. Even when he spoilt on himself I would wash his clothing and linen.”*
*(66-year-old)*


*“I am the mother; it was my child.”*
*(76-year-old)*

They performed a variety of caring activities and took that as their parental obligation and responsibility despite the physical effect on themselves.


*“I said to myself that this is my child, there is nothing I can do. I cannot ask someone else from outside…, Even when he died, he died in my arms. Even when he was dying, he screamed for me during the night. He called me and I thought that I was dreaming…, I slept with my bedroom door open.”*
*(64-year-old)*


*I even stopped putting him in the bed because I could not help him off the bed. He was sleeping on a mattress on the floor.”*
*(76-year-old)*

Caring for their sick children caused some of the participants to make significant sacrifices, including of the work that provided them with the money to support their entire families. For some of the participants, when their children became very ill, they had to stop work to care for them.


*“I requested my employers to go home and look after her even when that other lady was there; I also needed to be with her.”*
*(70-year-old)*

#### 3.2.3. Inadequate of Resources

The illness made the sick adult children unproductive and unable to work, and they thus needed financial care. This put a strain on the participants, who had to use their old-age pension money in order to take care of their sick adult children. In some cases, the participants had to split the inadequate resources in order to take care of their sick adult children even when they did not reside in the same household. They shouldered the responsibility for all in their children’s households, including the grandchildren, using their pension money in the process.


*“I had money problems … There was this time when I had no money and his father had no money too. We receive our pension money on the first day of the month, and in the middle of the month, our money is finished. If my son got sick, where would I get money? Eish … I would just have stress and the blood pressure will just go up.”*
*(60-year-old)*


*“When I get there, I just ask him what it that I can help him with. Maybe he slept without eating last night or maybe he forgot to take pills… I would buy him mealie meal. I am also supporting four (4) other parent adult children with my pension money.”*
*(Age unknown)*

Part of the caring role of the participants was to accompany their adult children to health facilities for follow up and treatment refill. The participants had to rely on hired private transport to get to the health facilities, as ambulance services are inadequate in rural settings. The need for transportation was frequent, because people with AIDS-related illnesses are frequently sick as the disease progresses. The frequent need for transport put a strain on the finances of the elderly parents especially because they had limited resources.


*He asked me to accompany him to the clinic. He wanted me to go everywhere with him. He knew he was dying.”*
*(64-year-old)*


*“He went to the clinic regularly to collect his treatment.”*
(72-year-old)


*I was the one who collected pills for my child from the clinic and would buy special porridge that would make my child active.”*
*(Age unknown)*

#### 3.2.4. Emotional Toil

AIDS was devastating to the participants, who watched their children become very ill and helpless, lose their appetites, vomit and develop diarrhoea. In some families, the illness affected more than one adult child at different periods, and for some at the same time, affecting the adult children, the adult grandchildren and the infants.


*“I was surprised, seeing that my second-born child was also sick. Eish ... I was shocked that both my adult children had HIV, the dangerous disease. I wondered if my children would survive ... but eventually they got strong and were willing to take treatment well.”*
*(76-year-old)*


*“Yes, he told me… the sister told me that she is also HIV positive and I asked myself how come two of my children can have this virus.”*
*(66-year-old)*

The participants experienced lots of pain from the illness of their adult children. The pain was not just from the fact of having their loved ones ill but from the knowledge and perception that HIV was a deadly disease. Nevertheless, they had to keep strong for their children and the rest of the family that was also suffering from emotional pain.


*“This illness is traumatising, painful in an indescribable way.”*
*(67-year-old)*


*Yoh…, I had a lot of stress, but now all that has passed. Now I can even advise other people but I did struggle a lot. I suffered.”*
*(69-year-old)*


*“Even when I felt pain, I was not supposed to show her that I felt pain. I was supposed to say, it is ok.”*
*(70-year-old)*

The loss of their adult children to AIDS affected the participants emotionally. Losing a child that had completed school felt like losing a beacon of hope.


*“Sometimes I would sit down and ask myself lots of questions such as: why must this happen to me and my child when I expected so much from him … He was educated, he had a law degree and I was waiting …, and hoped that he would work and be able to help me. He did not work at all, when he finished at the university, he got sick and this made me to be very heartbroken …”*
*(69-year-old)*

Worry and anxiety affected the participants if their sick children were unable to eat and yet had to take ART. They understood food to be the source of strength and a basic requirement before taking treatment as well as gaining energy for survival. Their children’s vomiting after a meal was a cause for concern for the participants, who consequently became despondent.


*“I had to force him to eat … I was worried because he would not get better if he did not eat ... When he ate, he would become alright.”*
*(60-year-old)*

Some of the participants experienced aggression from their adult children when they were caring for them. Some of the children got confused because in its advanced state, AIDS can affect the brain.


*“She was aggressive, very aggressive.”*
*(Age unknown)*


*What can I do if a person is like that because sometimes a person like that swears at you for no reason, they are short tempered over trivial things.”*
*(63-year-old)*

For some of the participants, taking care of their sick adult children was characterised by their experiences of the agony of HIV and seeing their loved ones suffering and gradually perishing in front of their eyes.


*“As their elderly parent I saw them getting sick. The other one I saw her coughing and getting skinny and always sleeping with no energy.”*
*(65-year-old)*


*“I was heartbroken because I know this HIV disease; if you do not take care of yourself, it will kill you.”*
*(69-year-old)*

#### 3.2.5. Sense of Fulfilment

Although the participants experienced challenges in caring for their children, they also experienced a sense of fulfilment for having done so. They felt fulfilled even when caring for their children took a toll of their physical strength. They felt they had to fulfil their parental role.


*“I would do what she could not do for herself…, I would sing for her and read the word of God. I would feed, bath, left her to sleep, and went outside. The time for meals was 12h00. I would come again and stay with my child, chatted, played, joked with her and even kissed her. She would look at me, I would sing for her until she fell asleep and I would leave her.”*
*(65-year-old)*

The prolonged periods of care and togetherness made them feel closely attached to their children. When their children’s health improved and they were able to undertake their daily activities and live a meaningful life, they felt fulfilled.


*“Yes, my child was finally better and was able to attend clinic, walking on her own…. the child is alive … Yes, she eventually got a grant … She got better.”*
*(73-year-old)*


*“Oh they are both girls, they are still alive…. They now have been ill for 10 years, and they are taking pills.”*
*(66-year-old)*

#### 3.2.6. Receiving Support for Care

Most of the participants were the sole caregivers for their sick children. Nevertheless, some reported that they received support from their families, neighbours, friends and church members.


*“The family really supported me, especially my children. That is why I am still alive until now as you can see. You see, if you do not have faith you will not survive and you will not talk about it the way I talk about it.”*
*(66-year-old)*


*“My family used to send us money whenever I tell them that we were short of something. They would send us money because we used to put my child in diapers.”*
*(64-year-old)*


*“There is only one neighbour that supported me; she would take her to her place for a while.”*
*(73-year-old)*

#### 3.2.7. Stigma and Discrimination

The data revealed nondisclosure and secrecy around HIV/AIDS. Most of the sick adult children delayed disclosure of their HIV status to their elderly parents. When the participants eventually learned about the child’s HIV status, they kept it secret to protect their children and the family from stigma and discrimination.


*“It is difficult… HIV is a disease that has to be kept secret. It is difficult to tell a person that your child has HIV.”*
*(63-year-old)*


*Mom, I am going to fetch my treatment at the clinic they said that I am HIV- positive’ so she asked me not to tell anyone.”*
*(70-year-old)*

Some of the participants related illnesses experienced stigma from close family members and community, while others did not. The few participants who disclosed the adult child’s status to family and others reported incidences of discrimination and open gossip about the sick adult child by the family members and others.


*They do things like, if the person can sit on a chair they will not sit on that chair; they will not even use the plate they have used; actually they do not want to see themselves next to the HIV-positive person.”*
*(65 year-old)*


*“My neighbours came and supported me but because they knew that he is HIV-positive, they would not come inside to see him.”*
*(69-year-old)*

However, most of the participants indicated that the problem of stigma is not as great as when the HIV and AIDS epidemic started. Almost all (24) indicated that their family did not experience any discrimination; however, none of them had disclosed the HIV status of their sick child to anyone outside of the immediate family.


*“The community does not discriminate them, they discriminate themselves.”*
*(66-year-old)*


*“I have not seen any discrimination. They treat them well.”*
*(73-year-old)*


*“No, I never experienced such, oh no.” *
*(70-year-old)*

## 4. Discussion

Caring for the sick adult child happened in the context of multigenerational living arrangements. However, living in these households did not relieve the older persons from the role of caring for their sick adult children, as other members of the household did not support them. The setting of the current study was an impoverished rural area, which is characterised by unemployment, daily commuting to cities and the migration of the working class to the big cities and mines, leaving the elderly alone. Therefore, most of these persons were the sole caregivers for their adult children, despite the fact that some of them (12 out of 31) were over 70 years of age. This study confirms the findings of previous studies conducted in sub-Saharan Africa, which indicate that older women are responsible for providing care to their adult children living with HIV/AIDS [11,30].

The study found that the participants could not be precise in terms of the duration of the illness and care; less than half indicated that the duration of care ranged from 6 to 11 months. While the data show that over half of the older persons cared for their children for a long time (more than 1 year), some learned about the HIV-positive status of their adult children only at the terminal stage. Some of the adult children returned home only at the terminal stage of their illness, which explains why almost all (24) the children cared for were reported to have died. None of the old persons was caring for a sick adult child during data collection; those whose children survived reported that they were on antiretroviral treatment and were living positively.

When the adult children got sick from AIDS-related illnesses and the illness advanced, the elderly parents felt obliged to take care of and ensure the well-being of their adult children. The sense of obligation arose from the realisation that if they did not do so, no one else would. In view of the fact that they were the sole care providers for their children, the demand for care took a toll on their physical and emotional health. Previous studies conducted in South Africa have reported similar findings [19,30,31].

Providing physical care is labour-intensive and puts physical pressure on older people, who are already weak by virtue of their age and health conditions. Almost all (27) the participants suffered from hypertension with comorbidities such as diabetes mellitus. They have to exert themselves when performing caring activities such as bathing and turning their adult children, carrying them and washing their soiled linen. Similar to what previous studies have reported, caring for adult children has a long-term effect on the elderly carers. Some of them experienced backache, leg pain and chest pain due to their caring activities as well as a reduced quality of life [32]. They maintain that the role of caring was overwhelming and stressful both physically and emotionally and often, they did not have the energy or strength to lift the adult child they cared for [9,21]. They became worn out because of the long hours for which they watched the child, their inability to sleep due to their anxiety and their fear that the sleeping child might never wake up. Moreover, they are at the stage of their lives when they would expect to be taken care of by their children, rather than the other way around. Physical infirmity brought about by their ailments and fatigue affected their performance of their newly acquired role of caring, including overlooking their own health [21].

The participants reported that the demanding nature of taking care of a sick child consumed all their time, and they were thus unable to undertake any other personal, family, church, community or work-related activities. Other studies have made similar observations that the demanding nature of caring for their sick children prevents older persons from going about the business of their daily lives [9,33]. As the illness progressed and the demands for care increased, some of the older persons sacrificed financial gain by quitting their employment for a while or for good, in order to take care of their sick child. Similar findings were reported elsewhere [33]. In other studies, it was observed that taking care of sick adult children led to the social exclusion of the carers, as they spent all their time looking after the children [13,19]. Care-giving activities also clashed with agricultural and community activities that the older persons were involved in, which resulted in a loss of income and food insecurity [9]. Consequently, the older persons performed their caring role under resource-constrained circumstances, which had challenged their commitment to care. As already said, older persons in rural areas are faced with a high rate of unemployment of their adult children, even before the children are taken ill, and they have to depend on their pensions to provide care. Schatz et al. found that pensions for the elderly play a very important role in female-headed households affected by HIV and AIDS [34].

Although the participants felt obliged to care for their adult children, they experienced various challenges, such as the emotional trauma that arose from taking care of adult children who were sick for a very long time. As in previous studies, the parents affected by HIV/AIDS experienced psychological problems such as feelings of anxiety, sadness, depression and fear [9,19]. In the current study and others [21,35], emotional trauma also arose from watching their children’s condition progressively deteriorate, as well as the knowledge that their children would never recover. AIDS illnesses are debilitating and often fatal, and thus, observing the progressive suffering of their children was emotionally traumatic to the parents [30,36].

The elderly carers lamented the inability of their sick children to take their medication as required, their deteriorating nutritional status, their gradually increasing physical weakness and their inevitable death. Some other studies have also alluded to the emotional trauma caused by the worsening condition of the sick child [9].

They also experienced anxiety brought about by the fear of the outcome of the illness, which in most cases was imminent death, as they knew that AIDS had killed a number of people in their communities and or families. The fact that AIDS-related illnesses drag on for a long time and are incapacitating brought about anxiety and distress for the carers, who often felt helpless and hopeless to cope with the situation at hand. Hopelessness and anxiety were also brought about by the sick individuals’ reluctance to eat, which the carers understood as having a negative effect on their nutritional status, their ability to take medication and the chances of their health improving.

Anxiety was occasioned also by the perception of HIV and AIDS as a dreadful disease that gives rise to stigma and discrimination. Negative feelings such as stigma and discrimination directed at those with AIDS-related illnesses are extended to their parents and the entire family [13]. The fear of stigma discouraged disclosure and compelled the older persons to keep the HIV status of their children secret. They felt obliged to keep the status of their children secret despite their belief that the levels of stigma and discrimination in the community had decreased. Most were requested by their sick children to keep the status secret. Secrecy is used to protect the HIV status of individuals when the environment is not conducive to disclosure [37].

Despite the hardship and challenges of taking care of the sick children, some of the older persons derived a sense of fulfilment from caring for their children and performed their role with zeal, passion and dedication. This dedication for caring resulted in their being reluctant to have their children admitted to hospital, even when they were at the terminal stage of their lives. It also motivated them to commit to caring for them, should other children become sick with AIDS in future.

This has implications for intervention and training, given the natural role adopted by older persons in families. If this is their attitude towards caring, it is important that they be given the necessary support and training to perform this role. There is no information on formal programmes targeting the family’s caregiving role, which is common in most sub-Saharan African countries. Therefore, the elderly assume the natural role of caring without the necessary basic knowledge and skills required for caring for people with AIDS-related illnesses. Healthcare workers in PHC facilities should raise awareness and educate communities about the availability of community-based structures such as the home-based care, palliative care and ward-based community services. The need to build the capacity of older persons to enhance their role without or with minimal physical and psychological effects has been reported in other studies [38,39]. The older persons expressed the need for food supplements and gloves as forms of support to enable them to take care of their sick children.

Consistent with other studies, the older persons reported that caring for their adult children provided them with a sense of purpose, pride and joy [40]. They developed a strong bond with and attachment to their children during the caring period. They become very close as they lived together, even though their children were very sick and became close care and treatment partners. Other studies reported on the parental bond that made the elderly parents willing to sacrifice and meet their parental responsibility to care for their children regardless of the adverse consequences [4,11,35].

## 5. Conclusions

This paper has found that as much as the elderly undertake the caring role with diligence, it is never without challenges, some of which arise from their lack of the resources and support with which to undertake the caring role. As a result, they are burdened with having to care for adult children with AIDS-related illness. Fatigue arising from the hard work of physically caring for their sick adult children day and night, as well as the lack of rest, add to the physical burden on the older persons.

Although there is a decrease in the number of AIDS-related deaths due to the benefits of ART, older persons will continue to carry the burden of caring for people with AIDS-related illnesses due to the increase in the number of new infections in South Africa. The study highlights the plight of the elderly and the tremendous role they play in the management of HIV and AIDS in their communities with or without support from the family, the community and the health department. Therefore, it is essential that the National Department of Health recognise that older persons are major providers of care for those who are sick and for orphaned grandchildren and ensure that policies and programmes address their needs. Although the International Plan of Action on Ageing urges governments to recognise the fact that HIV and AIDS epidemics affect older persons in multiple ways [41], the reality on the ground, especially in rural areas, tells a different story in terms of their plight. There is a need to target them and involve them in HIV and AIDS programmes as educators and learners. Their experience and wisdom would surely contribute positively to the benefit of individuals, families and communities and would assist in addressing HIV prevention.

## 6. Limitations

The study is subject to some limitations. The researcher chose a rural setting in which to conduct the study, and this limits the generalisability of the study findings to the urban population of the elderly, as their experiences of caring for adult children might be different, given the different setting. The study intentionally targeted people that are 60 years and above, as this is the cut-off age to define older people in South Africa in terms of the receipt of a pension. It is noted that there are other parents younger than 60 years who cared for adult children but were excluded from the study.

## Figures and Tables

**Table 1 ijerph-16-03162-t001:** Sociodemographic, health and caring characteristics of older persons.

Variables	Categories	Frequency	Percentage
Age group	62–69	19	61.3
	70–82	12	38.7
Education attainment	No formal schooling	15	48.4
	Primary education	13	41.9
	Secondary education	3	9.7
Duration of care for sick adults	Less than a year	15	48.4
	1–2 years	6	19.4
	>2 years	7	25.8
	Do not remember	2	6.4
Age of sick adults cared for	20–30 years	10	32.3
	31–35 years	5	16.1
	36–52 years	8	25.8
	Do not know	8	25.8
Outcome of care for sick adults	Died	24	80.0
	Alive	6	20.0
Sex of adult child with AIDS-related illness	Female	21	70
	Male	9	30
Reasons for visiting the clinic	Not feeling well	4	12.9
	To collect medication	27	87.1
Self-reported health status	Good health	12	38.7
	Poor health	19	61.3
Effects of caring on health	Did not affect health	13	41.9
	Affected health	18	58.1
Taking chronic medication	No	3	9.7
	Yes	28	90.3
Taking medication for hypertension	No	4	12.9
	Yes	27	87.1
Taking medication for diabetes mellitus	No	19	61.3
	Yes	12	38.7
Hospital admission in past 12 months	No	19	61.3
	Yes	12	38.7
Family experienced discrimination	No	24	77.4
	Yes	7	22.6

**Table 2 ijerph-16-03162-t002:** Summary of themes and subthemes.

Theme	Subtheme
HIV knowledge	
Learning about the HIV status of the child	Reaction to knowing
Perceptions about disclosure	Perceived reasons for nondisclosure
	Stigma and discrimination
The challenges to care	The physical effects of providing care
	Emotional toll
	Endurance
	Limited resources
	Sense of fulfilment
	Support for care
Risk of infection	
Perceived role in HIV prevention	HIV education
	Encouraging adherence to ART
	Encouraging HIV testing
	Promoting condom use

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
