# Peer review of "The Burden of Caring: An Exploratory Study of the Older Persons Caring for Adult Children with AIDS-Related Illnesses in Rural Communities in South Africa"

_ijerph, 2019, doi:10.3390/ijerph16173162_

Round 1

Reviewer 1 Report

Congratulations, you write a good and relevant paper. This is an important topic of studies whose results have to be taken in account by the National Health Service. The situation is dramatic for both the PLWHIV and their caregivers. I includd in the PDF some suggestions to improve the quality of the article.

Author Response

Did you ask the participants about the time of caregiving? It's important to study the burden

Response: Thanks for your comments on our manuscript. We added a table to provide detail information on the socio-demographic, health, and the caring characteristics of the participants. We had asked the participants this question and they indicated the duration of care, which ranged from 6 months to over 24 months. This is also addressed in the document (line 333-349).

2) To understand the reaction of the family caregivers, it is important to know if there are cormobities and the level of dependence of the PLHIV

Response: Thanks for this comment, again in table 1, we show that almost all the participants (27) were on treatment for hypertension and twelve of those with hypertension were talking medication for diabetes mellitus. We also report that caring for adult children has a long-term effect on the elderly carers (line 387-392) who had to do everything for the sick child. .

Please identify the participants with a letter, e.eg, I1, I2, I3 or INT1, INT2, INT3

Response: We apologize that we cannot identify the participants using a number; we saved the interviews using the names of the participants and we removed the names when we wrote the manuscript. The participants are older people and it would be culturally insensitive for the interviewers not to refer to them as Mrs, Mama, or Gogo so and so during the interviews..

Well, why not Self-care instead of Endurance? It's more common: self care, selfcare deficits, deendence on selfcare.

Response: Thanks for the comment; we understood endurance as the ability to endure an unpleasant or difficult process or situation without giving way. This is the essence of the theme where the older persons had no choice but to provide care even we caring affected them negatively. We therefore, would like to keep the theme.

It is very important. For families with a low income and weak support from the NHS, the situation can be dramatic.

Thanks for the comment (in line 411-414) we discuss the consequences of quitting employment in order to take care of their sick child.

It should be interesting to add here a theme objectively related to burden / fatigue, e.g. because of the time spent with the relative, they don't have enough time for themselves, stressed between caring for the relative and trying to meet other responsibilities,

The relative currently affects the relationship with family members or friends in a negative way that they don't have as much privacy as they would like because of your relative that their social life has suffered because caring for the relative, lost control of their lives since the relative's illness. It seems that authors have information about these issues

Response: The theme “The physical effects of providing care” address issue on fatigue for performing labour intensive tasks. In this theme, we show that reduced physical strength resulted in their inability to look after their households and affected their caring role. This is also highlighted in the discussion (line 383-384).

Reviewer 2 Report

Overall the study needs some major revisions especially sampling, results and discussion.

Specific Comments

1). Line 16 – “Thematic approach” vs “mixed method approach” in the methodology section (Lines 88 – 98) is rather confusing.

2).  Line 110 – These sample/participants presented at the hospital to collect chronic medication. It is important to highlight their health status at this point and how it factors into this specific study. This is significant in terms of their role as well as perspectives to quality of life (their own and that of their children). The participants are not a homogeneous group and these differences are key for the reader to know how it reflects in the results.

3). Line 128 – How about inclusion of a question/information specifically asking about any unmet needs, and ‘Other experiences’ that could be a result of what their roles such as the impact of social exclusion on their lives and health? The author(s) cannot make assumptions such does not exist or not relevant to the overall picture.

4). The study does not make it clear if the participants are still caring for the sick children or some have died and they are relying on the memories of how it was then. The reader gets an idea it is both but the study should make clear how many of these had lost their adult sick children and how many were still caring for their sick children who were alive. This is reflected in where the language could be used with present tense and in some cases past tense.

5). Line 157 – Write “5” in word (Five) since it is the start of a sentence.

6). Line 161 – Necessary to state all the themes and then explain why one is chosen. Having left out this presents a major challenge because the results are quite sketchy and barely justify the discussion and conclusions made. The title of the study is ‘Burden of caring” but not well reflected in the very narrow theme/results presented here.

7). Would be significant to know what the ages of the ‘adult children cared for’ are. This also highlights some of the cultural limitations of care across genders when the child is adult. The study should explain how the caregivers navigated such cultural issues or how it challenged their roles. African communities are tied in very strong cultural values that cannot simply be ignored and some can pose major psychological trauma.

8). Line – 261- States that ‘recovered’ from HIV/AIDS. How? Did you mean improved? There is still no cure for the disease and so ‘recovered’ can be confusing to the reader.

9). Line 268 – Sub theme 3.25 appears too brief for any meaningful separation as a category. Could be just an extension any of the other themes. Just one statement from a single participant is not enough to generalize.

10). Line 281 – Most participants were over 70 years contradicts what is stated in Line 156 (Most participants were between ages 62 – 66 years).

11). Line 284 – Talks about the aspect of the ‘duration of the illness’ yet the results did not show any question was asked or theme that highlighted this issue.

12). Line 312 – How did the caregivers own state of health (hearing loss or difficulties, mobility issues and chronic pain, failing eye-sight, memory lapses, dealing with their chronic diseases especially multi-comorbidities etc) impact their roles or management of their own health?

13). Line 336 - How did they navigate things like stigma (this is not reflected in the results yet somehow discussed), or fear of possible infection to themselves? Do they take it for granted it cannot happen? Also discuss the actual and/or potential effects of social exclusion for these caregivers.

14). Line 345 – Specify what kind of ‘support’ or training is needed.

15). Line 377 – HelpAge International has worked hard and made major strides in highlighting the plight of the older adults’ roles in fighting this epidemic, the term “forgotten group” may be changing several decades since the start of this disease.

Author Response

Overall, the study needs some major revisions especially sampling, results and discussion.

Specific Comments

1).Line 16 –“Thematic approach” vs “mixed method approach” in the methodology section (Lines 88 – 98) is rather confusing.

Response: We have rephrased the statement on data analysis to show what approach was used for the qualitative and quantitative data in the abstract and in the relevant section in the document (line 140-141, and line 150).

2). Line 110 – These sample/participants presented at the hospital to collect chronic medication. It is important to highlight their health status at this point and how it factors into this specific study. This is significant in terms of their role as well as perspectives to quality of life (their own and that of their children). The participants are not a homogeneous group and these differences are key for the reader to know how it reflects in the results.

Response: Thanks for the comment; we added a table to provide detail information on the socio-demographic, health, and the caring characteristics of the participants. The participants who participated in the study were visiting the facilities for chronic medication refill, none were critically ill but. Although we could not ascertain how their health status was before their caring role, we asked their perceptions of how caring affected their health.  

3). Line 128 – How about inclusion of a question/information specifically asking about any unmet needs, and ‘Other experiences’ that could be a result of what their roles such as the impact of social exclusion on their lives and health? The author(s) cannot make assumptions such does not exist or not relevant to the overall picture.

Response: Thanks for the comment; the study asked about the participants’ experiences of rejections from family members and or community and their opinions on how to deal with discrimination, it was an omission from our part not to include these questions. We added these questions (line 130-131).

Concerning unmet needs, the older persons expressed the need for food, pills, clothes and gloves to enable them to take care of their children. They also wished to be educated on HIV, and how to take care of their adult children with HIV/AIDS (line 458-459).

4). The study does not make it clear if the participants are still caring for the sick children or some have died and they are relying on the memories of how it was then. The reader gets an idea it is both but the study should make clear how many of these had lost their adult sick children and how many were still caring for their sick children who were alive. This is reflected in where the language could be used with present tense and in some cases past tense.

Response; none of the participants were still caring for their adults children, we provided information in table one and (line 378-380). We addressed the language to reflect the past tense.

5). Line 157 – Write “5” in word (Five) since it is the start of a sentence. This has been corrected

6). Line 161 – Necessary to state all the themes and then explain why one is chosen. Having left out this presents a major challenge because the results are quite sketchy and barely justify the discussion and conclusions made. The title of the study is ‘Burden of caring” but not well reflected in the very narrow theme/results presented here.

Response: Thanks for the comment: we added two themes “3.2.3. Inadequate resources” and “3.2.7. Stigma and discrimination” and we added additional quotations to some of the themes to address the concern about the sketchy results. The new quotes are in blue font colour. We also added a table with all the themes and provided an explanation for selecting the burden of care.

We hope that the manuscript will not be too long with these additions.

7). Would be significant to know what the ages of the ‘adult children cared for’ are. This also highlights some of the cultural limitations of care across genders when the child is adult. The study should explain how the caregivers navigated such cultural issues or how it challenged their roles. African communities are tied in very strong cultural values that cannot simply be ignored and some can pose major psychological trauma.

Response: the age of the adult children cared for is presented in table 1. We did not find that caring for a male adult child was a cultural barrier. One of the mothers had this to say, I bathed him, dressed him, and did everything for him. I asked him to stop being afraid of me because he can see that he is unable. I am his mother.” (75 year-old). Another said, “No there was no one and especially when he had running stomach I was the one who was helping him” (72 year-old).

8). Line – 261- States that ‘recovered’ from HIV/AIDS. How? Did you mean improved? There is still no cure for the disease and so ‘recovered’ can be confusing to the reader.

Response: we changed recovered’ from HIV/AIDS to when their children’s health improved and they were able to undertake their daily activities

9). Line 268 – Sub theme 3.25 appears too brief for any meaningful separation as a category. Could be just an extension any of the other themes. Just one statement from a single participant is not enough to generalize.

Response: subtheme 3.2.5 deleted,

10). Line 281 – Most participants were over 70 years contradicts what is stated in Line 156 (Most participants were between ages 62 – 66 years).

Response: This mistake has been corrected and the statement now reads as; “despite the fact that some of them (12 out of 31) were over 70 years of age”(line 370).

11). Line 284 – Talks about the aspect of the ‘duration of the illness’ yet the results did not show any question was asked or theme that highlighted this issue.

Response: The duration of care is presented in table 1, in line 374-375 we indicate that the duration of care ranged from 6 to over 24 months.

12). Line 312 – How did the caregivers own state of health (hearing loss or difficulties, mobility issues and chronic pain, failing eye-sight, memory lapses, dealing with their chronic diseases especially multi-comorbidities etc.) impact their roles or management of their own health?

Response: We indicate that providing physical care is labour-intensive and puts physical pressure on older people, who are already weak by virtue of their age and health conditions. We also report that caring for adult children has a long-term effect on the elderly carers. Some of them experienced backache, leg pain and chest pain due to their caring activities as well as a reduced quality of life, and that often, they did not have the energy or strength to lift the adult child they cared for. We support this with literature (line 387-394). We also report that they showed commitment in taking care of them and took this to be their sole responsibility even if it meant ignoring their own health needs. .

13). Line 336 - How did they navigate things like stigma (this is not reflected in the results yet somehow discussed), or fear of possible infection to themselves? Do they take it for granted it cannot happen? Also discuss the actual and/or potential effects of social exclusion for these caregivers.

Response: we added a theme to address stigma and discrimination and show that because of the fear of stigma, the participant did not disclose the HIV status of their sick children to people outside of their close family members (line 437-443)

14). Line 345 – Specify what kind of ‘support’ or training is needed.

Response: we added information describing the type of training that should be provided to older persons (line 451-459)

15). Line 377 – HelpAge International has worked hard and made major strides in highlighting the plight of the older adults’ roles in fighting this epidemic, the term “forgotten group” may be changing several decades since the start of this disease.

Response: Thanks for suggesting this policy framework, we have integrated the recommendations from HelpAge in the conclusion and deleted the term “forgotten group” (line 478-484).